# Estimating Oil Sands Emissions using Horizontal Path-Integrated Column Measurements

Timothy G. Pernini[1], T. Scott Zaccheo[1], Jeremy Dobler[2], Nathan Blume[2]
[1]Atmospheric and Environmental Research, Inc, Lexington, Massachusetts, 02421, USA
[2]Spectral Sensor Solutions, LLC, Fort Wayne, Indiana, 46818, USA

*Correspondence to*: Timothy G. Pernini (tpernini@aer.com)

**Abstract.** Improved technologies and approaches to reliably measure and quantify fugitive greenhouse gas emissions from oil sands operations are needed to accurately assess emissions and develop mitigation strategies that minimize the cost-impact of future production. While several methods have been explored, the spatial and temporal heterogeneity of emissions from oil sand mines and tailings ponds suggests an ideal approach would continuously sample an area of interest with spatial and temporal resolution high enough to identify and apportion emissions to specific areas/locations within the measurement footprint. In this work we demonstrate a novel approach to estimating greenhouse gas emissions from oil sands tailings ponds and open-pit mines. The approach utilizes the GreenLITE™ gas concentration measurement system, which employs a laser absorption spectroscopy-based, open-path, integrated column measurement in conjunction with an inverse dispersion model to estimate methane ($CH_4$) emission rates from an oil sands facility located in the Athabasca region of Alberta, Canada. The system was deployed for extended periods of time in the summer of 2019 and spring of 2020. $CH_4$ emissions from a tailings pond were estimated to be 7.2 t/day for Jul-Oct 2019, and 5.1 t/day for Mar-Jul 2020. $CH_4$ emissions from an open-pit mine were estimated to be 24.6 t/day for Sep-Oct 2019. Descriptions of the measurement system, measurement campaigns, emission retrieval scheme, and emission results are provided.

## 1. Introduction

Oils sands are a natural combination of sand, water, clay, and bitumen – a viscous hydrocarbon mixture – that are a source of unconventional petroleum and can be refined to produce crude oil. The largest known deposits of oil sands exist in Canada and Venezuela, with lesser deposits in Kazakhstan and Russia [Tong, 2018]. Significant deposits of bitumen exist in the Canadian province of Alberta to include the Athabasca, Cold Lake, and Peace River regions [Vigrass, 1968; Mossop, 1980; Hubbard, 1999]. While all three regions are suitable for production using in-situ "drilling in place" methods, such as cyclic steam stimulation (CSS) or steam assisted gravity drainage (SAGD), the Athabasca region is particularly suited to surface mining due to the relatively shallow depth of bitumen deposits. After oil sands have been mined, the ore is mixed with hot water and chemical solvents to separate bitumen for extraction. The remaining components are called tailings and are typically stored in large, engineered dam systems called tailings ponds with the long-term goal of land reclamation [Nix, 1992; Hossner,

1992]. Tailings ponds are known emitters of greenhouse gases, volatile organic compounds, and other atmospheric pollutants [Englander, 2013; Burkus, 2014; Small, 2015; Bari, 2018]. Several factors can influence their emission rates, which include pond size, tailings discharge method/flow rate/location, tailings type/age, and local climatic conditions that include air temperature, wind, rain, and ice cover [Small, 2015]. A similar set of factors can influence mine emission rates, including mine size, local climactic conditions, and mining activities.

The quantification of fugitive greenhouse gas emissions from oil sands operations is needed to provide a better understanding of the underlying chemical and process-based mechanisms, and to provide estimates of annual emissions which enable the development, regulation, and enforcement of limits on total allowable emissions per site and monetary incentives that promote emissions reductions and carbon neutrality [AEP, 2019]. As such, improved technologies and approaches to reliably measure and quantify emissions are needed to effectively assess true emissions and develop mitigation strategies that minimize the cost-impact of future production. Since emissions from oil sands mines and tailings ponds are spatially heterogeneous and vary temporally, an ideal approach to measure and quantify emissions would continuously sample an area of interest with spatial and temporal resolution high enough to identify and apportion emissions to specific areas/locations within the measurement footprint. To date, several measurement techniques have been explored or implemented at oil sands sites, including flux chambers and aircraft mass balance approaches, as well as micrometeorological techniques that include eddy covariance (EC) instrumentation, flux-gradient (FG) observations, and inverse dispersion modeling (IDM).

Flux chambers [Klenbusch, 1986] have been traditionally employed to estimate emission rates in the Alberta oil sands, but typically measure a small area (~0.13 $m^2$) for a short duration (0.5-1 h). This approach does not account for variability in emissions over time, and many samples across an entire site of interest are needed to account for non-uniform emissions from heterogenous sources such as oil sands tailings ponds and mines [Small, 2015]. Furthermore, several studies have suggested that the flux chamber measurement technique itself may influence and bias the true emission [Gholson, 1991; Pumpanen, 2004; Wells, 2011]. Even for a homogeneous source, flux chamber results have been shown to be 50-124% of the true emission rate [Klenbusch, 1986]. In a comparison study conducted over a 5-week period at a tailings pond, [You, 2021b] found that flux chambers underestimated emissions by a factor of two when compared to those from EC, FG, and IDM micrometeorological approaches.

The EC technique [Burba, 2013] is a well-established and widely accepted approach to estimate emissions from area sources due to its ability to measure vertical flux directly and continuously [Denmead, 2008; Zhang, 2013; Podjrajsek, 2016; Kukka-Maaria, 2017]. A challenge of the EC approach in an oil sands environment is that the measurement footprint of an EC system depends largely on the height of the measurement, wind speed, wind direction, atmospheric stability, and surface roughness/terrain and will vary in size and location over time [Burba, 2013]. An EC upwind fetch distance is commonly tens to hundreds of meters, while oil sands mines and tailings ponds can span several square kilometers. Multiple perimeter EC

tower measurements would likely be needed to adequately sample the spatial extent of an oil sands mine or pond, and even then, large areas at the centers of these sites would likely not be accounted for. Furthermore, the EC technique commonly requires strategies to account for data gaps, which occur for a variety of reasons and can result in over a 50% data rejection rate [Vesala, 2008; Zhang, 2018].

Several aircraft mass-balance studies have been conducted to quantify emissions from oil and gas operations [Karion, 2013; Petron, 2014; Karion, 2015; Lavoie, 2015; Peischl, 2015; Peischl, 2016], and specifically at Alberta oil sands operations [Gordon, 2015; Baray, 2018; Liggio, 2019]. Flight patterns used to accommodate the mass balance approach are typically single-height transects downwind of the assumed emissions source, a single screen that uses several downwind transects at multiple heights, or a full box/polygon that surrounds the source. The single-height transect approach assumes a vertically well mixed boundary layer, such that the species concentration is constant from the surface to the boundary layer height. The single screen method interpolates measurements at multiple heights to form a vertical 2-D distribution of gas. The full box approach builds on the single screen method by surrounding the emission source on all sides, at multiple heights, and emission rate is derived by the total advective fluxes through the surrounding screen. In all aircraft mass-balance approaches a measure of background is required, often achieved by upwind flight transects or by the outer extremes of a downwind transect which are assumed to be unaffected by the emissions source. All approaches require the assumption of steady horizontal winds over the course of measurements. Other factors impacting the accuracy of emissions obtained with these approaches are related to the time required to complete measurement flight patterns, including non-stable planetary boundary layer (PBL) height, wind conditions, entrainment between free troposphere and PBL, and background concentration [Peischl, 2015; Peischl, 2016]. Furthermore, continuous measurements are not feasible with an aircraft approach, and measurements over several days and different months and seasons would be necessary to evaluate the variability and seasonality of emissions [Karion, 2013; Karion, 2015]. While mass balance approaches are suited to relatively large, heterogeneous areas of emission, their ability to allocate emissions to specific zones within an area is limited. Furthermore, aircraft campaigns tend to be costly to do with any regularity.

In addition to EC, two other micrometeorological approaches to estimating emissions from area sources are the FG method and IDM. The FG method [Meyers, 1996; Bolinius, 2016] has previously been applied at oil sands operations [You, 2021a; You, 2021b] and is based on employing concentration measurements at two or more heights to approximate a concentration gradient from which flux can be deduced. Any means of measuring concentration at multiple heights can be used with the FG approach, but the measurement technique chosen will dictate the effective measurement footprint, temporal resolution, and apportionment ability of emission estimates. Examples include point measurements along the vertical of a tower [Todd, 2007], EC measurements at multiple heights [You, 2021b], or open-path Fourier transform infrared (OP-FTIR) spectroscopy measurements at multiple heights [You, 2021a]. The footprint or measurement fetch of various measurement techniques can vary significantly, and any approach could, in theory, be set up for either short-term emission studies or long-term, continuous monitoring. Similar to FG, the IDM approach [Flesch, 1995] can utilize a variety of measurement techniques. The inverse-

dispersion technique employs an atmospheric dispersion model to quantify the theoretical emissions associated with a measured concentration, where the assumed emission source is typically upwind of the concentration measurement location. All IDM approaches fundamentally require a measure of background concentration and a measure of emission source concentration. And like FG, the footprint or measurement fetch of the chosen measurement techniques can vary significantly. The IDM method has been demonstrated in various applications with open-path, integrated column measurements [Flesch, 2004; Flesch, 2005a; Gao, 2008; You, 2021a; You, 2021b]. An open-path, integrated measurement has the potential to reduce error in the IDM method in that it provides a more comprehensive measure of the air parcel under investigation and is therefore less susceptible to localized variations within a dynamic emission plume [Flesch, 2004], as compared to a point measurement.

In this work we demonstrate a novel approach to estimating greenhouse gas emissions from oil sands tailings ponds and open-pit mines with potential for broader applicability to both wide-area, diffuse emission sources and applications requiring leak source identification and quantification. The approach utilizes the GreenLITE™ gas concentration measurement system, which employs a laser absorption spectroscopy-based, open-path, integrated column measurement in conjunction with an IDM to estimate methane emission rates from an oil sands tailings pond and an open-pit mine located in the Athabasca region of Alberta, Canada. Descriptions of the measurement system, measurement campaigns, emission retrieval scheme, and emission results are provided.

## 2. Measurement System

GreenLITE™ is a laser absorption-based gas measurement system that consists of one or more optical transceiver units, some number of retroreflectors arranged such that a clear line of sight exists between each transceiver and each reflector, and backend analytics that convert measured optical depth values to gas concentrations in near real-time and generate 2-D concentration distributions [Dobler, 2015; Dobler, 2017; Zaccheo, 2019]. A transceiver consists of a stationary climate-controlled equipment cabinet and an optical head that is mounted on a two-axis mechanical scanner. GreenLITE™ is unique in its implementation of intensity modulated continuous wave (IMCW) laser absorption spectroscopy (LAS). A GreenLITE™ transceiver is configured to measure a specific gas by precisely setting the wavelengths of two laser sources such that one is strongly absorbed by the gas of interest and the other is minimally absorbed by that gas. The laser wavelengths chosen for GreenLITE™ allow for operation over path lengths up to 5 km while remaining well below the eye-safety limit. The utility and advantages of an integrated long-path measurement used in conjunction with an IDM in an oil sands environment has recently been demonstrated [You, 2021a]. The IMCW approach simultaneously transmits both wavelengths through the atmosphere, allowing for the cancellation of common-mode noise such as scintillation. By intensity-modulating each wavelength with a unique waveform, the laser energy returned by the reflector to the transceiver can be separated into the individual wavelength components, and the differential absorption between the wavelengths can be determined. The IMCW technique makes GreenLITE™ nearly immune to the largest sources of noise in other long-path LAS systems (e.g., scintillation). The

differential absorption of these two wavelengths by the gas can be directly converted to optical depth, from which the concentration of the gas can be determined using a radiative transfer model [Clough, 2005; Rothman, 2009] in an iterative scheme [Dobler, 2015; Zaccheo, 2019]. To help ensure that the system measurement precision is maintained, data quality filters are applied in a conservative approach to remove measurements that may be biased due to low signal level (affected by electronic noise) or high signal level (overloading or clipping of amplifiers and analog-to-digital converters).

While GreenLITE™ may be used to measure the concentration over a single atmospheric path, the more common system configuration involves the transceiver scanning to multiple reflectors to measure an area. The optical head is pointed at each reflector for some period of time, measuring the path-integrated concentration of the target gas along the straight-line path ("chord") from the transceiver to the reflector. If two transceivers are arranged such that their measurement chords intersect one another, a 2-D reconstruction of the distribution of the gas concentration over an area that can span up to 25 $km^2$ can be obtained. These 2-D field estimates are based on the use of a sparse tomographic approach [Dobler, 2015; Dobler, 2017] that minimizes the error in the observed space between an analytical model of the field, composed of a set of background terms and N idealized models of dispersion-based plumes, and the observed chord values. Typically, N is a small number on the order of 4 or less and is limited by the number of chords (information elements) that can be used to solve for the background and plume parameters. The wind direction and speed are used to constrain the direction and strength of dispersion, and the chord intersect values aid in the first guess choice of parameters.

The analytics portion of the GreenLITE™ system utilizes cloud processing. The measured optical depth data are uploaded to a cloud-based processing, storage, and display framework in real time where concentrations and 2-D distributions of concentration and emission are computed. A web-based interface provides near-real-time display of the data and can be configured to provide alerts via email or SMS text messages when operator-defined conditions are detected. Prior to deployments at the Alberta oil sands, GreenLITE™ has previously been tested and deployed in several environments, including 6 months at a carbon capture and storage facility in Illinois [Dobler, 2017; Blakley, 2020], a full year in the urban core of Paris, France [Lian, 2019], and a week-long campaign at an oil and gas processing facility in Lacq, France [Watremez, 2018].

Weather data required to support the gas measurement campaigns described in the next section were acquired with local instrumentation. In 2019, local air temperature, pressure, and relative humidity were measured with a Davis Vantage Pro2 weather station[1], and wind speed and direction were measured with a Campbell Scientific CSAT3B 3-D sonic anemometer[2].

[1] https://www.davisinstruments.com/product_documents/weather/spec_sheets/6152_62_53_63_SS.pdf, last accessed March 2021.
[2] https://www.campbellsci.com/csat3, last accessed March 2021.

In 2020, equivalent measurements were made using a METER ATMOS 14 weather station[3] and METER ATMOS 22 sonic anemometer[4].

## 3. Measurement Campaigns

GreenLITE™ systems were deployed to the operational oil sands facility in the Athabasca region of Alberta, Canada – once in the summer/fall of 2019 and a second time in the spring of 2020. During the 2019 campaign a single system was installed at a tailings pond (57.34126° N, 111.903790° W), operating continuously from June through October 2019, and a second system was installed at a nearby open-pit mine and operated continuously for nearly a six-week period in September and October 2019. The system at the tailings pond was configured in a dual-gas, non-mapping mode as shown in Figure 1, with two transceivers collocated on the west side of the pond and each configured to measure a different gas – the transceiver denoted as T3 measured carbon dioxide ($CO_2$), and that denoted as T4 measured methane ($CH_4$). While $CO_2$ concentration measurements were used to estimate $CO_2$ emissions from the tailings pond, estimating $CO_2$ emissions requires accounting for relatively large biogenic contributions and necessitates a significantly more detailed analysis and discussion than could be addressed in this paper. Therefore, $CO_2$ emission results will be addressed in a future publication, and this paper will focus on $CH_4$ emissions. Six reflectors were placed around the pond, denoted by R01 through R06, and formed six measurement chords between transceivers and reflectors. The four chords formed by reflectors R02 through R05 crossed over some portion of the pond, while the chords formed by R01 and R06 served as background measurements, assuming predominant winds from the west. The chord lengths ranged from just over 1 km to 4.8 km. Also installed at the transceiver location were the weather station and sonic anemometer referenced in the previous section, which provided meteorological data used in the retrieval of gas concentration from optical depth and in the estimation of emissions from the pond. The objective of the 2019 measurement campaign at the tailings pond was to estimate $CH_4$ and $CO_2$ emissions over an extended time period.

[3] https://www.metergroup.com/environment/products/atmos-14/, last accessed March 2021.
[4] https://www.metergroup.com/environment/products/atmos-22-sonic-anemometer/, last accessed March 2021.

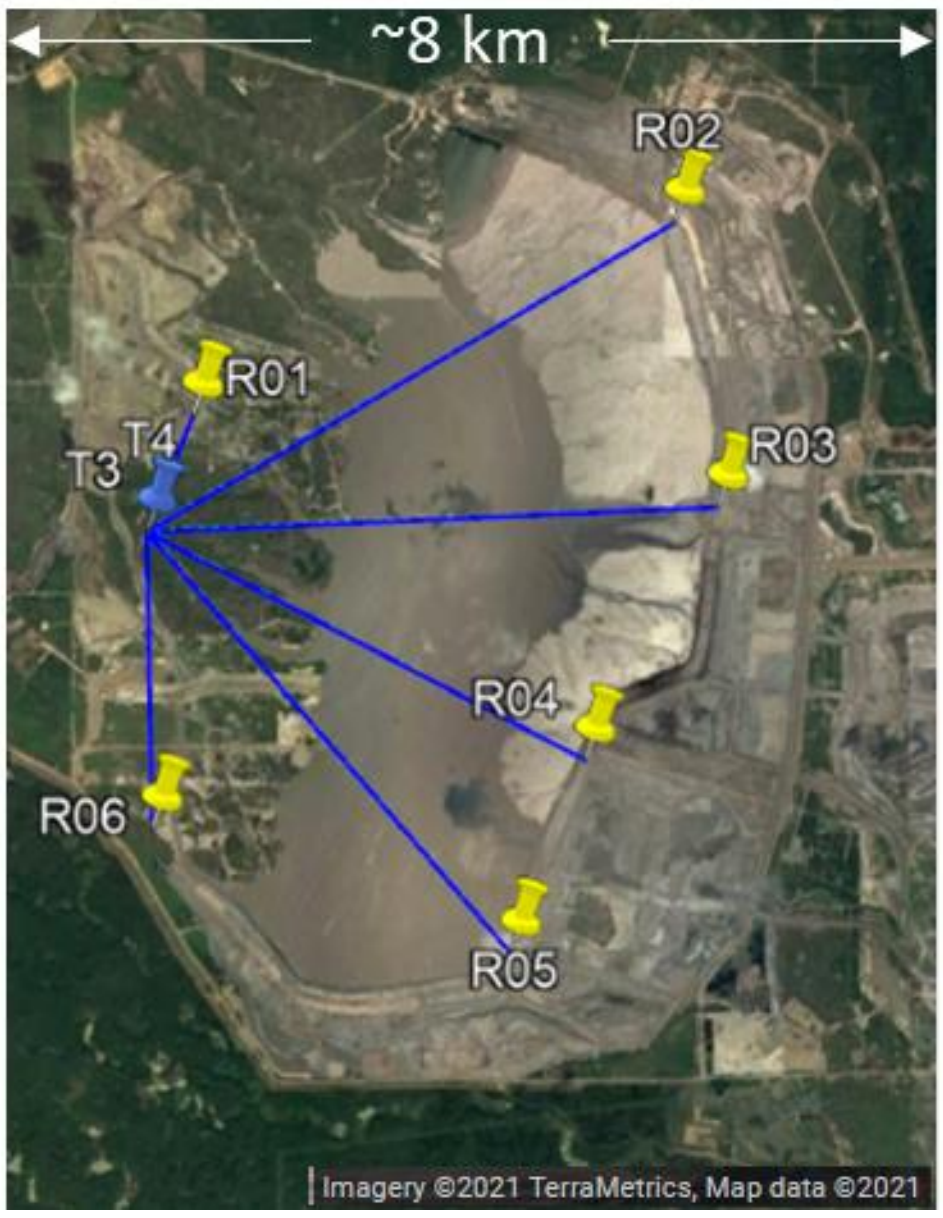

**Figure 1. GreenLITE™ system configuration at tailings pond.**

The system that was deployed to the open-pit mine (57.328044° N, 111.758565° W) in 2019 for six weeks was configured to measure $CH_4$ with the ability to generate 2-D concentration and emission maps. For mapping capability, two GreenLITE™ transceivers must be separated by a distance on the order of half the width of the area to be measured. In the deployed configuration at the mine, as shown in Figure 2, the transceivers – denoted T1 and T2 – were located 960 m apart on the north edge of the mine pit. Fifteen reflectors, denoted R01 clockwise through R15, were installed along the east, south, and west edges of the mine. Chord lengths ranged from 440 m to 2.4 km. The cross-hatched measurement chord pattern, shown in Figure 2, enabled the construction of 2-D maps of concentrations and emissions, based on a sparse tomography approach, that will be discussed later. Local surface weather data were used in the retrieval of gas concentration from optical depth and in the estimation of emissions from the mine. The objective of the 2019 measurement campaign at the mine was to estimate spatially resolved $CH_4$ emissions over an extended period.

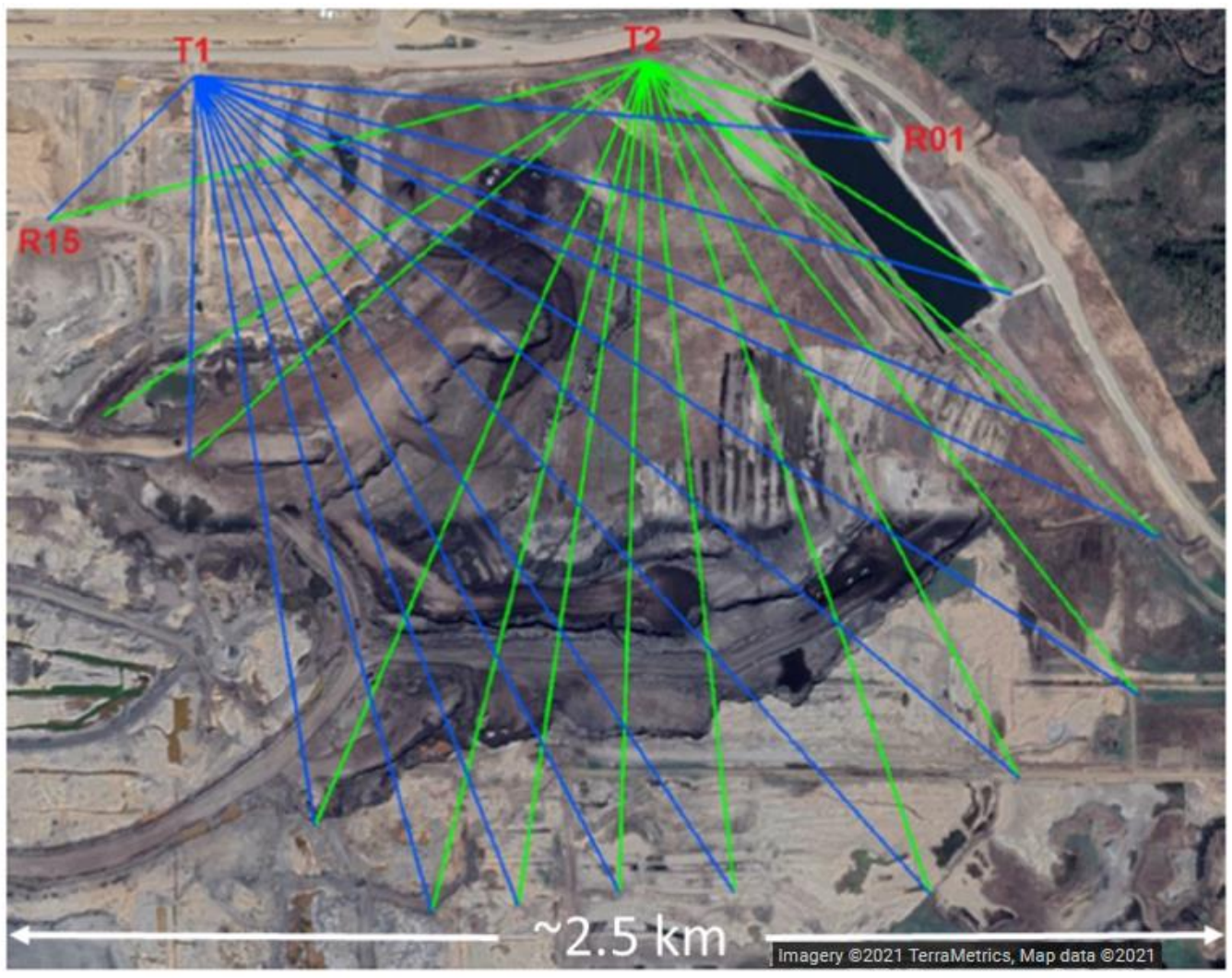

**Figure 2. GreenLITE™ system configuration at mine face.**

In 2020, the GreenLITE™ system was again installed at the tailings pond shown in Figure 1, and in nearly the same configuration. The objective of the 2020 measurement campaign at the tailings pond was to observe and quantify any potential enhancement in emission during the time of pond ice thaw and breakup, as an emission outgassing during this time had been postulated [Small, 2015].

## 4. Emission Estimation

The GreenLITE™ concentration measurements were combined with locally measured surface weather information, including air temperature, humidity, air pressure, wind speed, and wind direction; publicly available Numerical Weather Prediction (NWP) Rapid Refresh [Benjamin, 2016] upper-air model fields; and terrain information derived from the Canadian Digital Elevation Model (DEM)[5] to form the inputs to the Second-Order Closure Integrated Puff Model with Chemistry (SCICHEM) dispersion model [Chowdhury 2015] to estimate $CH_4$ emission rates. SCICHEM is based on the Second-Order Closure Integrated Puff (SCIPUFF) model [Sykes, 1986; Sykes, 1997] which was developed as a short-range dispersion model. A flow diagram depicting the emission estimation process is shown in Figure 4. Measured concentrations for each measurement chord shown in Figure 1 that passed over the pond, denoted by R02, R03, R04, and R05, were averaged on an hourly basis and background-corrected using the corresponding hourly-averaged concentration measurements from chords R01 and R06. Since R01 and R06 are most likely contaminated by pond emissions when winds have an easterly component, data were filtered for use in emissions estimates based on an acceptable wind range of 190° to 350° to ensure that concentration measurements taken

[5] NRCan (Natural Resources Canada), 2016. Canadian digital elevation model, 1945–2011. https://open.canada.ca/data/dataset/7f245e4d-76c2-4caa-951a-45d1d2051333, last accessed March 2021.

along chords R01 and R06 and used in background correction were upwind of the pond. The background chords formed by R01 and R06 were intentionally located on the west side of the pond since the prevailing winds for this site are out of the west.

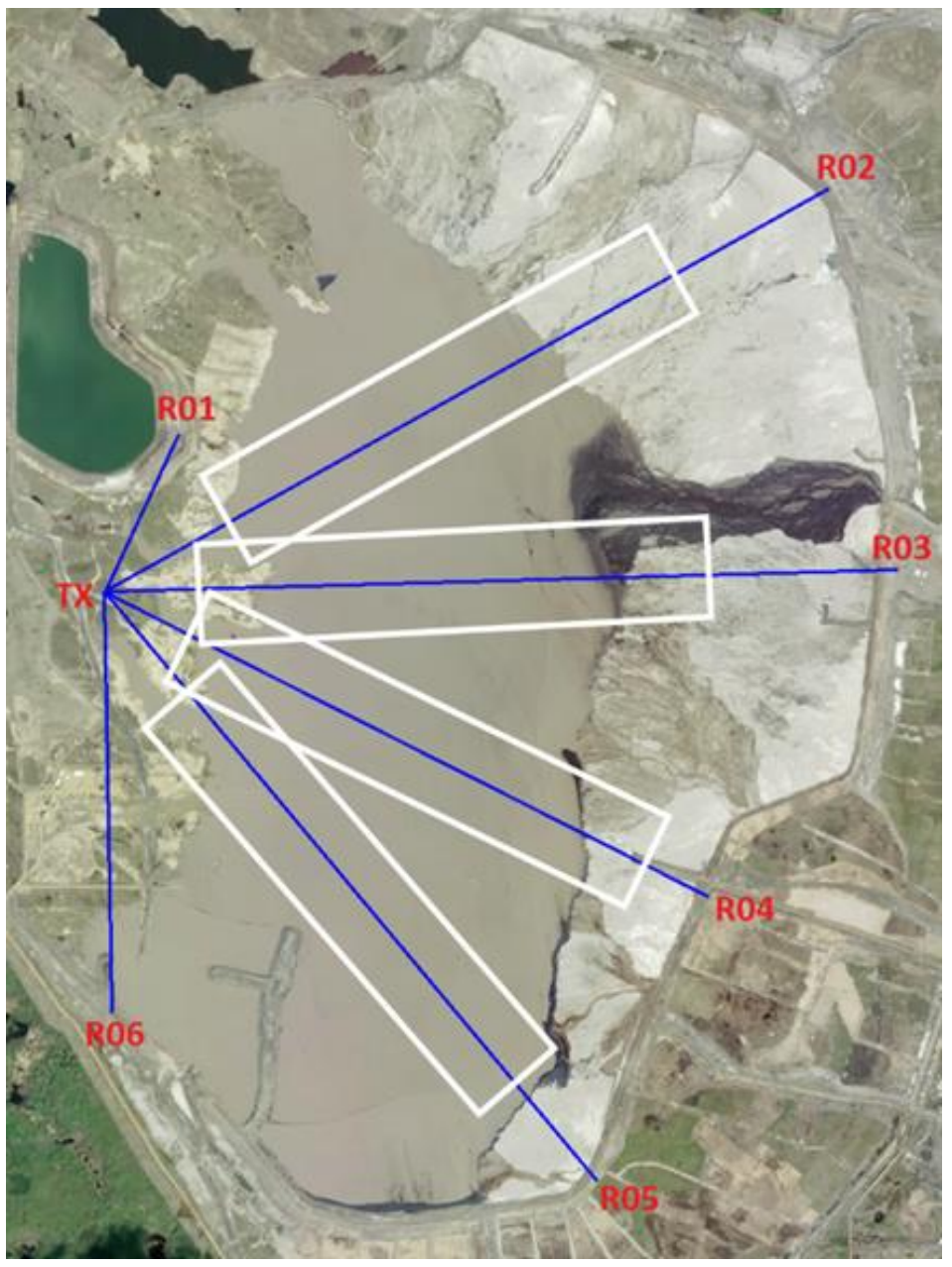

**Figure 3. Simulated area emission sources (white rectangles) used in SCICHEM modeling scheme for the tailings pond (image credit: CNRL, 2020).**

SCICHEM was run using continuous area source release scenarios as depicted by the notional white rectangles shown in Figure 3. While the GreenLITE™ concentration measurements that serve as input to the SCICHEM modeling framework are integrated measurements that span the pond and east beach, the SCICHEM model was limited to rectangular simulated release areas with constraints on release area dimensions. The simulated release areas for each chord measurement were centered at the midpoint of each respective chord. For any given set of chord measurements, SCICHEM was used to independently model the concentration in each rectangular box by computing a value given an initial guess at the associated emission rate. The initial guess is accounted for in the independent release scenario for each simulated release area. The differences between the hourly-averaged measured and modeled concentrations on a per-chord basis were then used in an iterative conjugate gradient scheme to adjust the emission rates until the modeled and measured concentrations matched within 0.0005 ppm. Once the values converged, the corresponding emission rates were recorded. Since the pond and east beach areas are considered as emission sources, the per-chord emission rates reported by SCICHEM (units of mass per unit time) were then normalized by the respective simulated release source areas of each chord to convert to flux (mass per unit time per unit area), averaged hourly for chords that transect the pond, and scaled by the estimated total pond (and east beach) area to provide hourly emission estimates for the entire site. This emission retrieval scheme was used to estimate $CH_4$ emission rates during both the 2019 and

2020 campaigns at the tailings pond for each hour that met the screening criteria for wind direction and included measurements that met a minimum signal-to-noise ratio (SNR) threshold.

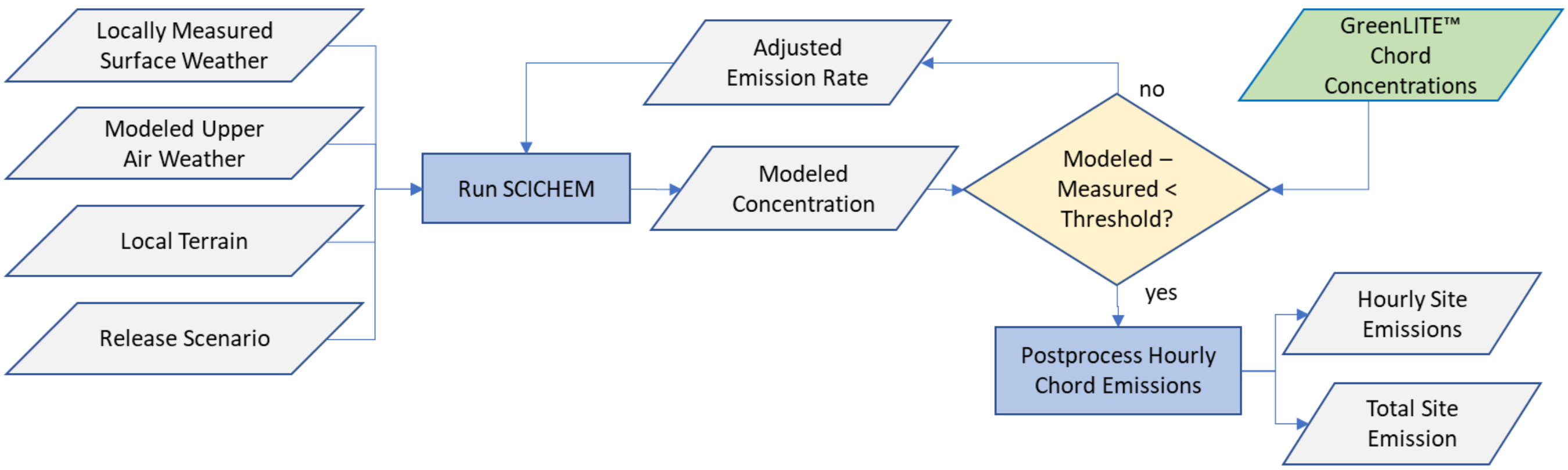

**Figure 4. Emission estimation flow diagram.**

A nearly identical approach was employed to estimate emissions from the open-pit mine in 2019. However, the simulated area release scenario used in the SCICHEM emission retrieval approach differed from that used at the tailings pond shown in Figure 3 since the configuration at the mine (Figure 2) allowed the development of 2-D maps of concentration at the approximate height of the plane formed by the measurement chords above the mine face. A plume-based model is used to predict the 2-D methane distribution associated with a collection of diffuse emission sources. An example plume-based 2-D concentration distribution estimate for the mine installation is shown on the left side of Figure 5. The plume-based 2-D maps are used as the basis to formulate a reconstruction scheme with rectangular basis functions that were employed during the 2019 mine campaign to provide a direct interface to standardized emission model frameworks, such as SCICHEM. In the case of the mine, geo-referenced rectangles and local topography from historical DEMs are used to describe sub-sections of the mine and surrounding areas. An example box-based concentration reconstruction is shown on the right side of Figure 5. The areal extent and concentration for each rectangle are directly integrated into the simulated release scenarios employed in the SCICHEM emission retrieval scheme. These boxes/rectangles serve the same purpose as the white rectangles shown in Figure 3 for the pond. In the case of the mine, the ambient $CH_4$ background concentration required by the dispersion modeling scheme was determined based on the box having the lowest concentration relative to all other boxes in a given 2-D distribution. In this manner, emissions were estimated for all time periods that had a 2-D reconstruction, irrespective of wind direction, which was a constraint on the single-transceiver setup at the pond.

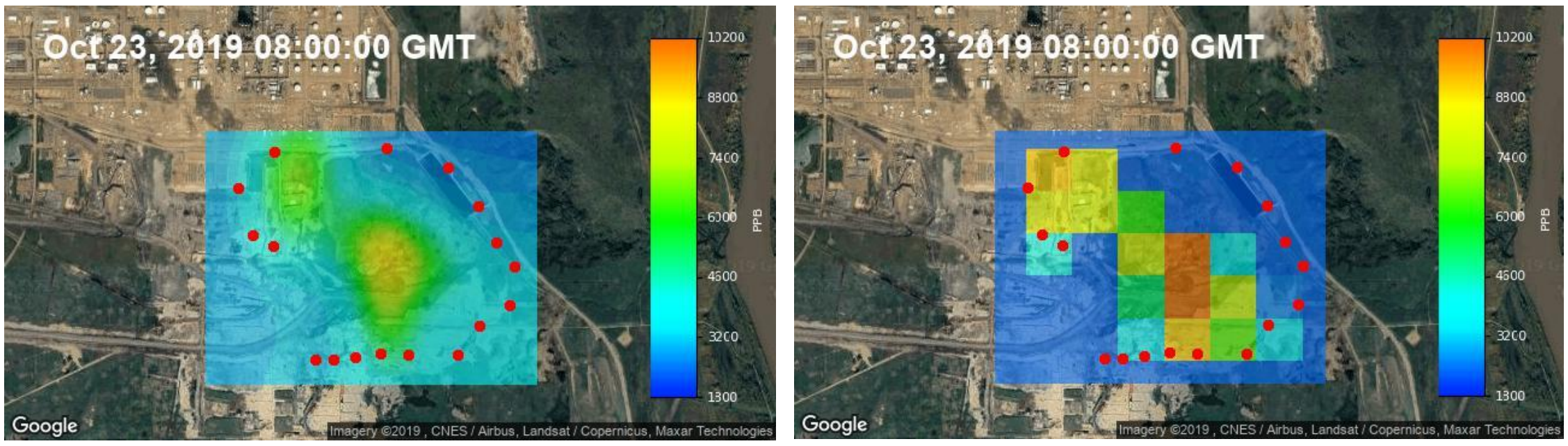

**Figure 5. Example plume-based (left) and corresponding box-based (right) methane concentration reconstructions at the open-pit mine.**

## 5. Results

### 5.1. Tailings Pond Emissions

$CH_4$ emission rates computed for the tailings pond are shown in Figure 6 for both the summer 2019 and spring 2020 measurement campaigns. Gray data points represent hourly emission rates, and blue data points denote the two-day moving average. Average daily emissions were 7.2 metric tons/day during the Jul-Oct 2019 period and 5.1 ± 2.9[6] metric tons/day during the Mar-Jul 2020 period. Accounting for the estimated area of the pond and east beach (17.7 $km^2$), which is also treated as an emission source, these seasonal emission rates scale to annual fluxes of 1.48 t/ha/yr and 1.05 t/ha/yr, respectively. As can be seen in Figure 6, emissions were higher and more variable in summer 2019 versus spring 2020. Several drivers may have contributed to this behavior. For example, the pond ice surface was frozen through a portion of the spring campaign, partially capping emissions from a large portion of the pond surface. The temporal variability in the results shown in Figure 6 also indicates the potential for significant biases in annually reported emissions that are based on periodic measurements versus an extended or continuous measurement approach.

[6] Uncertainty analysis provided in Section 5.3.
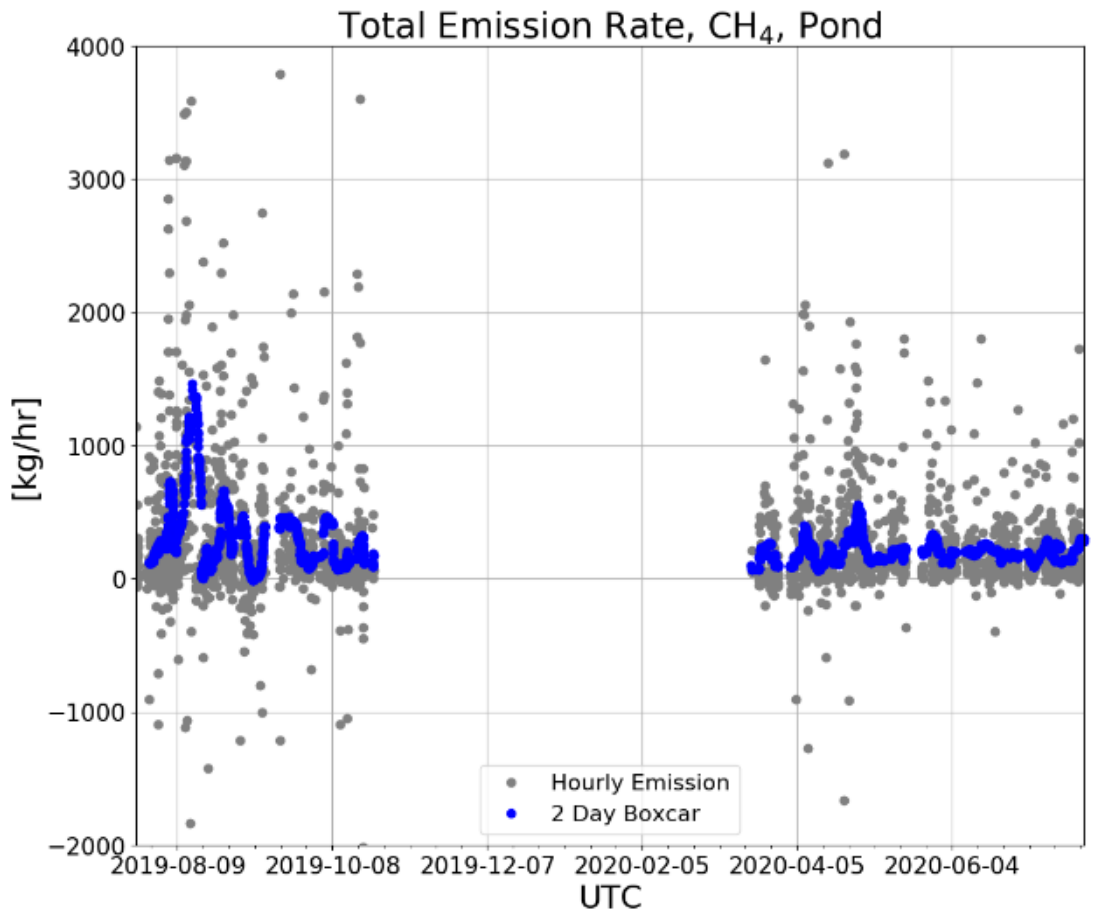

**Figure 6. Tailings pond methane emissions.**

As previously mentioned, a key motivation for the spring 2020 campaign was to measure and quantify any enhancement in emissions during the time period of pond ice breakup. First, to identify the period of pond ice breakup, local air temperature was studied as an indicator of ice thawing and eventual breakup. Figure 7 shows daily high, average, and low air temperature as measured at the pond site during early spring of 2020. Daily high temperature was consistently above freezing beginning the second week of April, and daily average temperature was consistently above freezing beginning the third week of April.

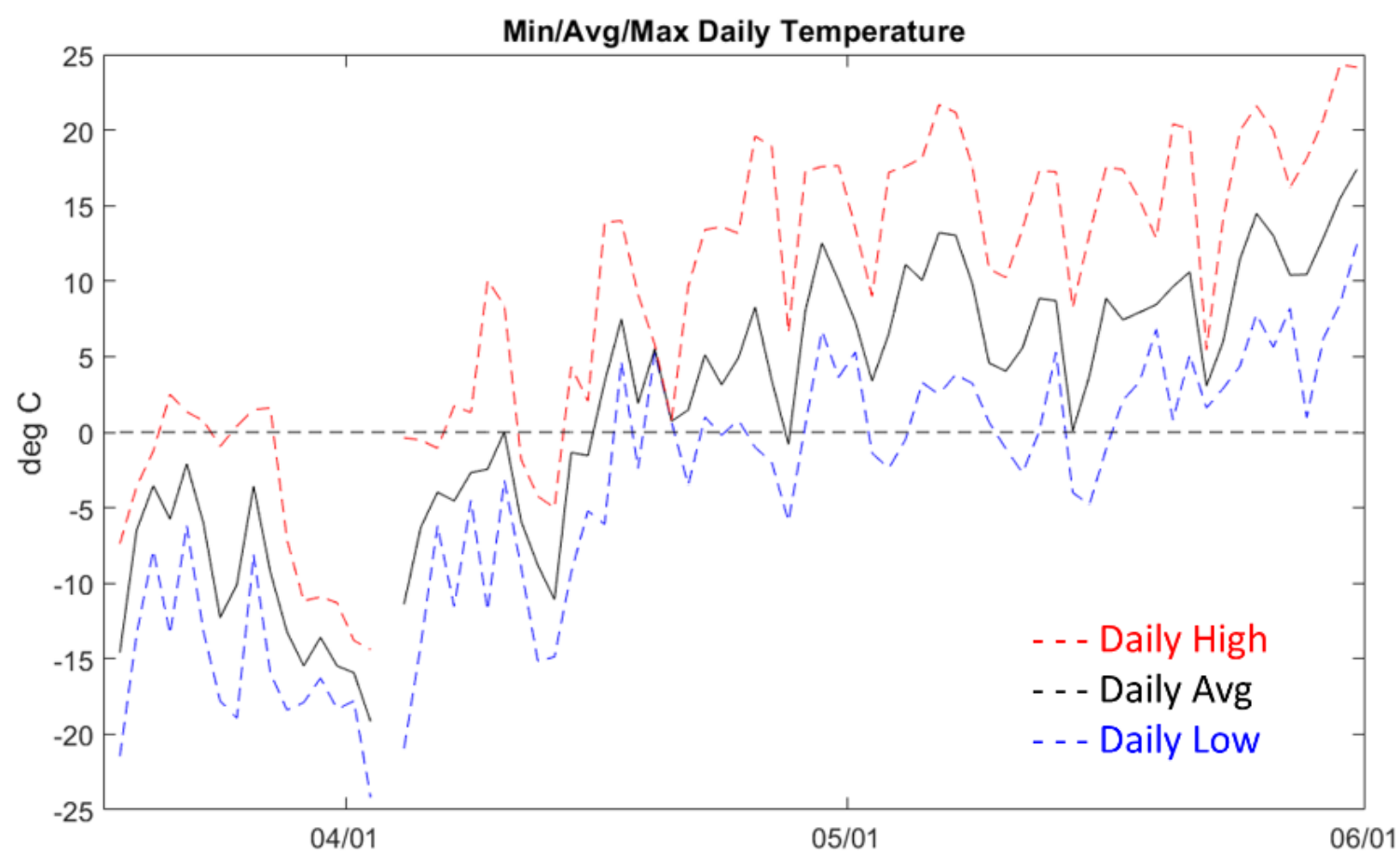

**Figure 7. Daily high, average, and low air temperature at tailings pond site during spring 2020.**

Furthermore, while the exact freezing/melting points of the tailings pond and nearby Athabasca River may vary due to differences in composition and dynamics, the local water level of the river was studied as an approximate indicator of regional ice thaw and breakup. Figure 8 shows the river water level through the last two weeks of April 2020, and strongly implies river ice breakup during the last week of April.

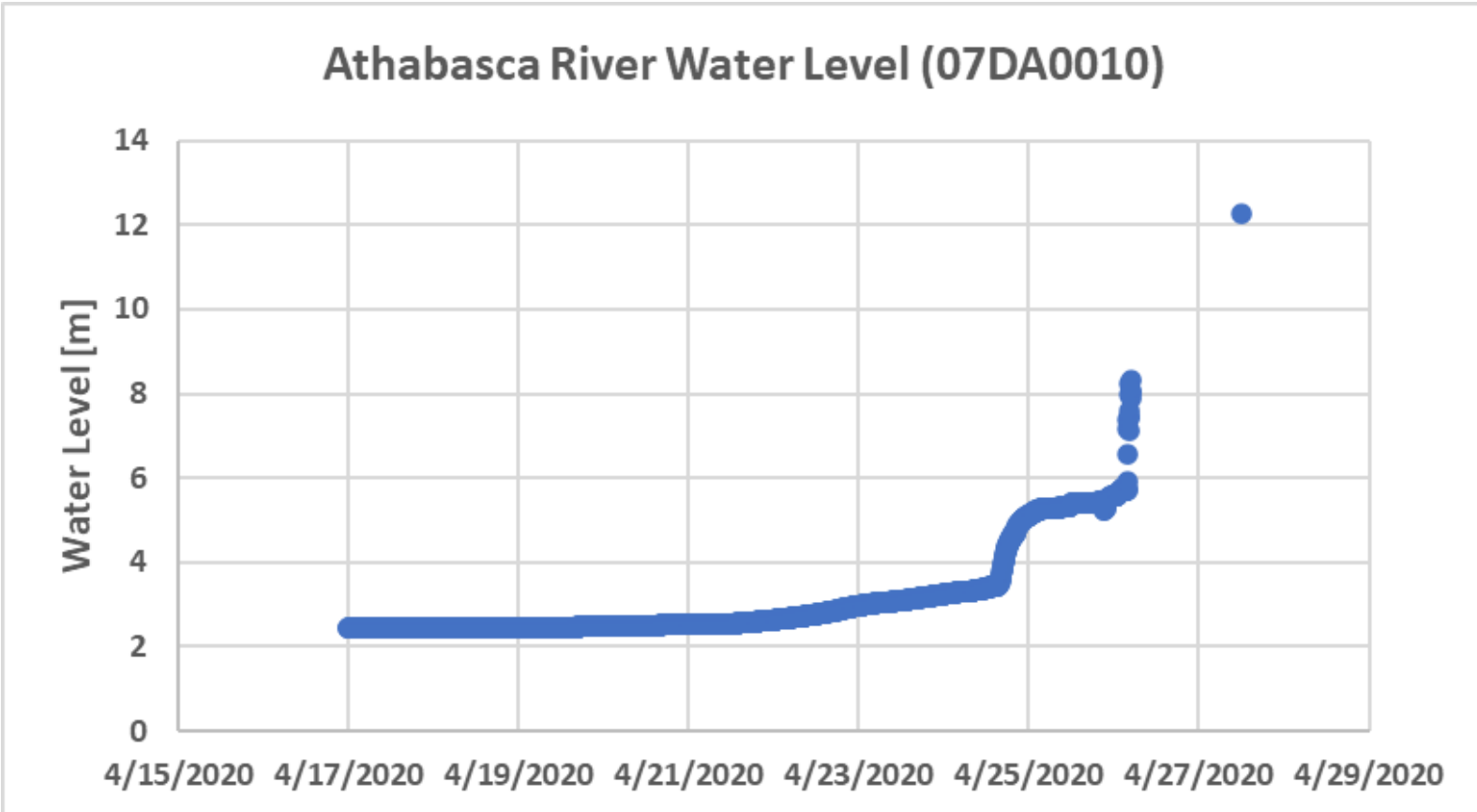

**Figure 8. Athabasca River water level[7].**

Figure 9 shows $CH_4$ pond emission through the 2020 campaign along with the two-day moving average of local air temperature (green) for reference. Two emission enhancements are discernible that may be associated with ice breakup. The first occurs during the second week of April, which coincides with daily high air temperatures consistently above freezing. The second and higher magnitude enhancement occurs during the last week of April, which overlaps with daily average air temperature consistently above freezing and is close in time with the assumed ice breakup of the Athabasca River. The late-April enhancement peaked at 739 kg/hr, based on the rolling two-day average, on April 28, 2020, and was 4 times the median hourly rolling average emission computed over the course of the 2020 campaign.

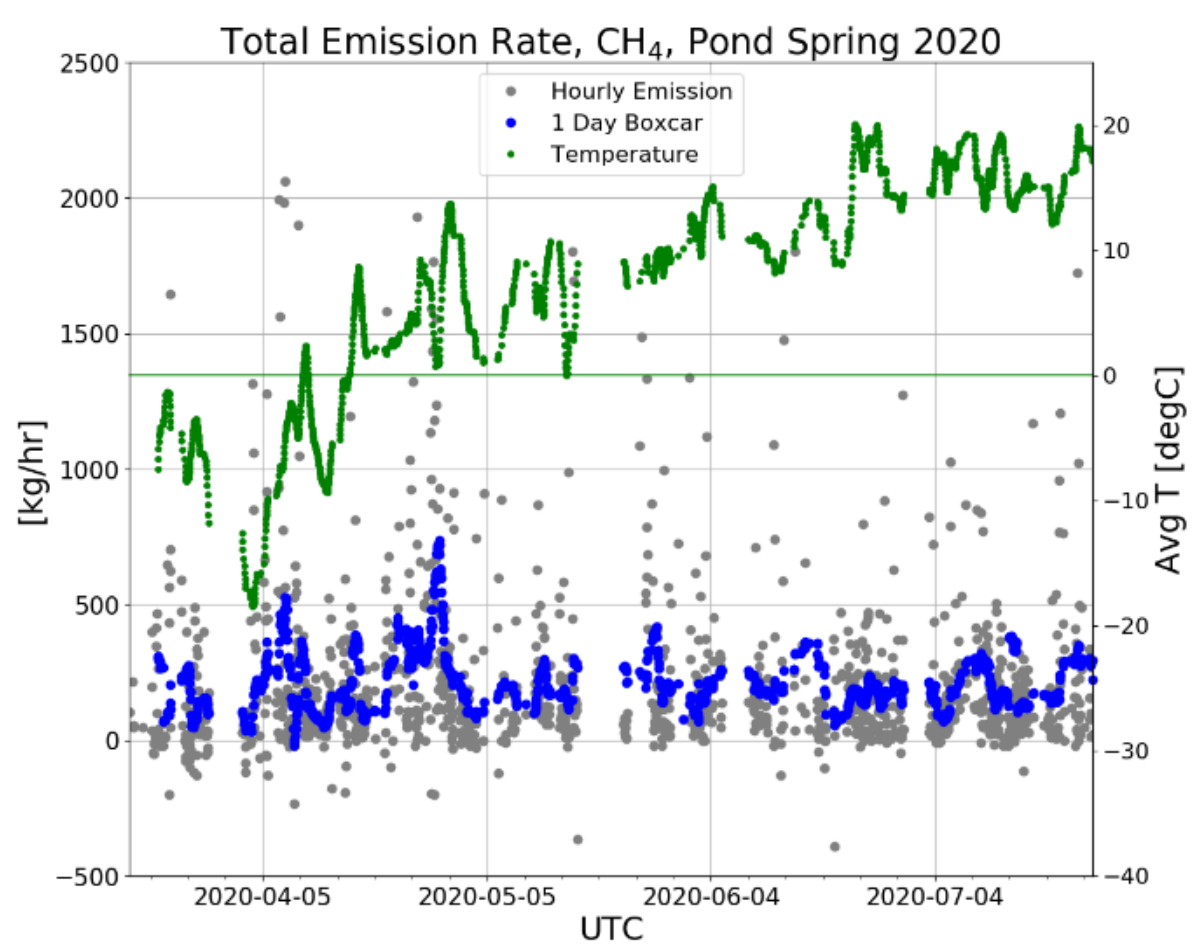

**Figure 9. Hourly emission (gray) and two-day moving average (blue) of $CH_4$ emission from tailings pond. Two-day moving average of local air temperature (green) corresponds to the right y-axis.**

[7] Government of Canada, Water Office, https://wateroffice.ec.gc.ca/, last access: March 2021
A diurnal pattern in $CH_4$ *emission* from the tailings pond was not discerned in an hour-of-day analysis of both summer 2019 and spring 2020 emission results, in contrast to [Zhang, 2018] which reported 2.8 times higher $CH_4$ emission at night versus day based on EC measurements over a 13-day period at an Athabasca tailings pond in June of 2012. However, a diurnal pattern was observed not only in measured $CH_4$ *concentration* over the pond, similar to that reported by [Zhang, 2018], but also in the background *concentration* as seen in Figure 10. The figure shows median hour-of-day concentration for the summer 2019 (left) and spring 2020 (right) measurement periods. The diurnal patterns seen in Figure 10 are indicative of the daily planetary boundary cycle which compresses the near-earth atmosphere at night. Additionally, the fact that both pond and background concentrations loosely track one another throughout this pattern is a good indication that the background measurements are not biased by local $CH_4$ sources.

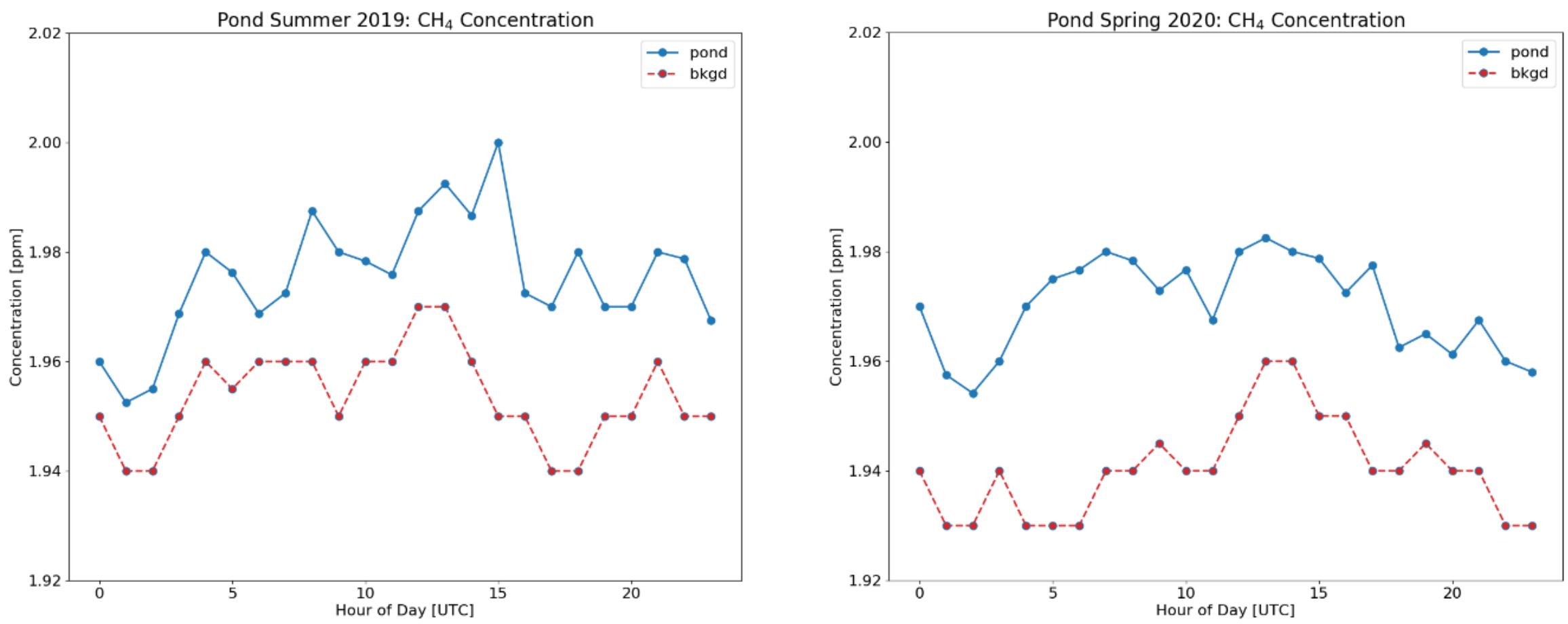

**Figure 10. Tailings pond and background $CH_4$ median hour-of-day concentration for summer 2019 (left) and spring 2020 (right).**

Several $CH_4$ emission studies have been conducted at the same tailings pond over the last decade. Results from many of these studies are summarized in Table 1. The measurement and emission estimation approaches covered in this table include flux chamber, EC, multiple point measurements with the WindTrax[8] IDM, multiple point measurements with the CALPUFF[9] IDM, and multiple open-path, integrated measurements with the SCICHEM IDM (GreenLITE™). The emission values shown in Table 1 provide a qualitative comparison of several emission estimation techniques from multiple seasons over many years. As such, conclusions drawn from direct comparisons should be made with caution for several reasons, most of

[8] http://www.thunderbeachscientific.com/, last accessed Mar 2021.
[9] http://www.src.com/, last accessed Mar 2021.

which have already been discussed. First, production at the site has increased over the past decade[10 11 12], which, in theory, would result in larger quantities of tailings and higher $CH_4$ emissions over this time. The data in Table 1 are also plotted in Figure 11 on a monthly basis and show a trend of increasing emission that may be indicative of the increase in production from the oil sands site, in addition to improved emission measurement techniques. Second, tailings pond $CH_4$ emissions have been shown to have a seasonal dependency. Lastly, the measurement footprint represented by each approach listed in Table 1 varies significantly, and tailings pond emissions are spatially heterogenous.

[10] CNRL Horizon 2010 oil sands production, https://www.cnrl.com/upload/media_element/369/02/0106_horizon-oil-sands-production.pdf, last access: March, 2021.
[11] CNRL 2019 year end results, https://www.cnrl.com/upload/media_element/1281/02/0305_q419-front-end.pdf, last access: March 2021.
[12] Canada's Energy Future 2017 Supplement: Oil Sands Production, https://www.cer-rec.gc.ca/en/data-analysis/canada-energy-future/2017-oilsands/index.html, last access: March 2021.

**Table 1. Historical tailings pond emission studies [AECOM, 2021]**

| Method | Year | Sampling Period | Pond $CH_4$ Emission (t/yr) |
|---|---|---|---|
| Flux Chamber | 2012 | Late Aug | 959 |
| | 2013 | Mid Oct | 187 |
| | 2014 | Early Aug | 727 |
| | 2016 | Aug-Sep | 1799 |
| | 2017 | Early Aug | 1905 |
| Eddy Covariance | 2017 | Mid Aug | 1945 |
| | 2018 | Jun-Aug | 2415 |
| | 2019 | Mar-Apr | 1867 |
| | 2019 | May-Jun | 3139 |
| | 2019 | Jul-Aug | 1862 |
| WindTrax-IDM | 2015 | Sep-Oct | 409 |
| | 2016 | Aug-Sep | 649 |
| | 2018 | Apr-May | 8500 |
| | 2019 | Feb-Mar | 6453 |
| | 2019 | Jul-Aug | 6383 |
| | 2019 | Oct-Nov | 1154 |
| CALPUFF-IDM | 2015 | Sep-Oct | 2712 |
| | 2016 | Aug-Sep | 1052 |
| | 2017 | Mid Aug | 1592 |
| | 2018 | Apr-May | 9873 |
| | 2019 | Apr | 2520 |
| | 2019 | Aug | 2550 |
| | 2019 | Sep | 1073 |
| GreenLITE™ | 2019 | Jul-Oct | 5001 |
| | 2020 | Mar-May | 1935 |

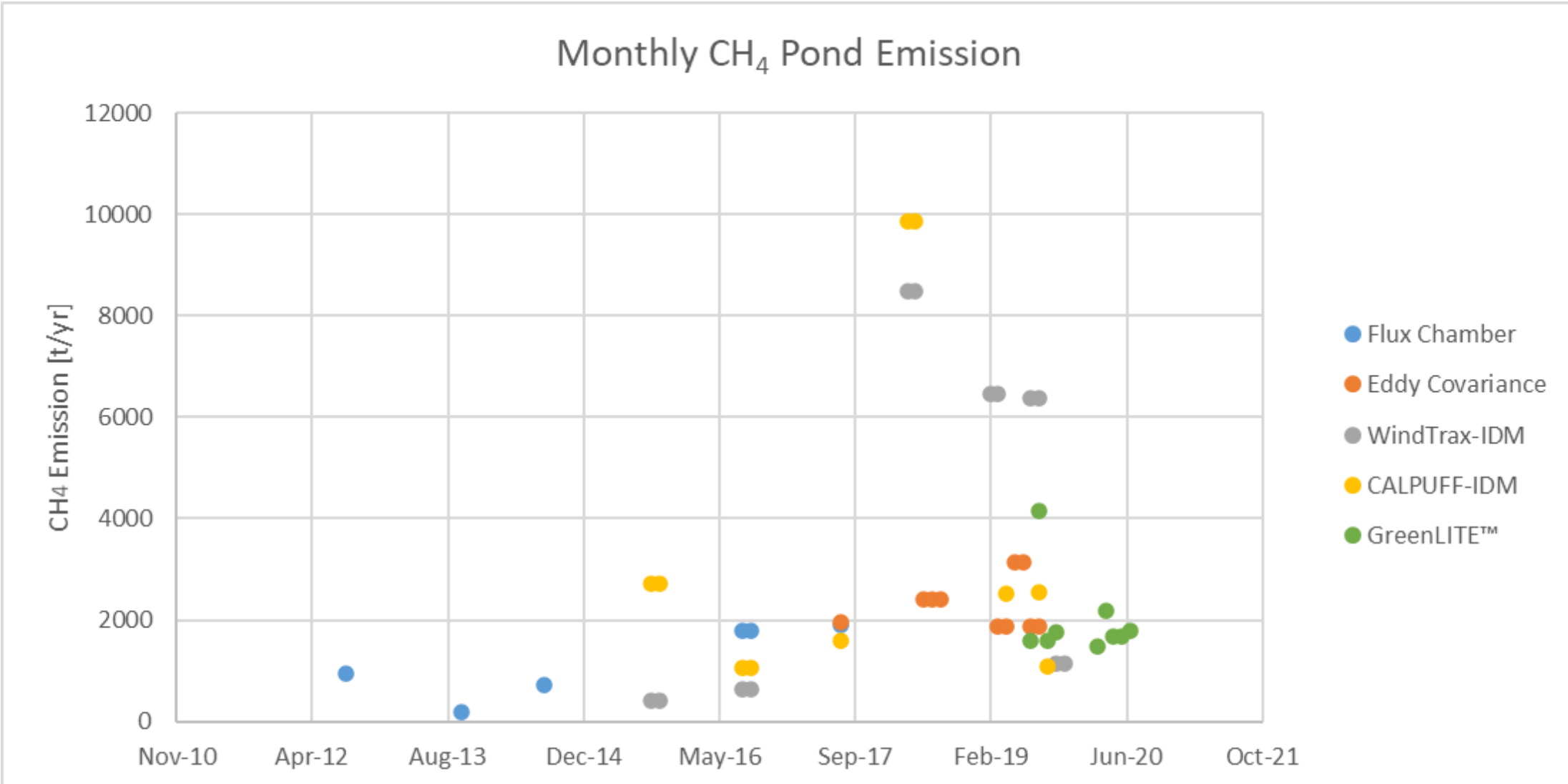

**Figure 11. Historical tailings pond emission studies.**

Figure 12 shows reported monthly bitumen production at the oil sands site during 2019 (left) and 2020 (right), with $CH_4$ tailings pond emission as computed with GreenLITE™ overplotted. Since tailings produced will vary as a function of bitumen mined, so too will $CH_4$ pond emissions be expected to vary. As can be seen in the figure, $CH_4$ emissions trend well with bitumen production except for the first month of each respective GreenLITE™ measurement campaign. Not coincidentally, both of the pond measurement campaigns began mid-month in July 2019 and March 2020, respectively, causing those months to be undersampled and further emphasizing the importance of continuous or longer-term measurement. Of 744 hours in July, only 84 hours were sampled near the end of the month during the summer 2019 campaign. Of 744 hours in March, only 108 hours were sampled near the end of the month during the spring 2020 campaign.

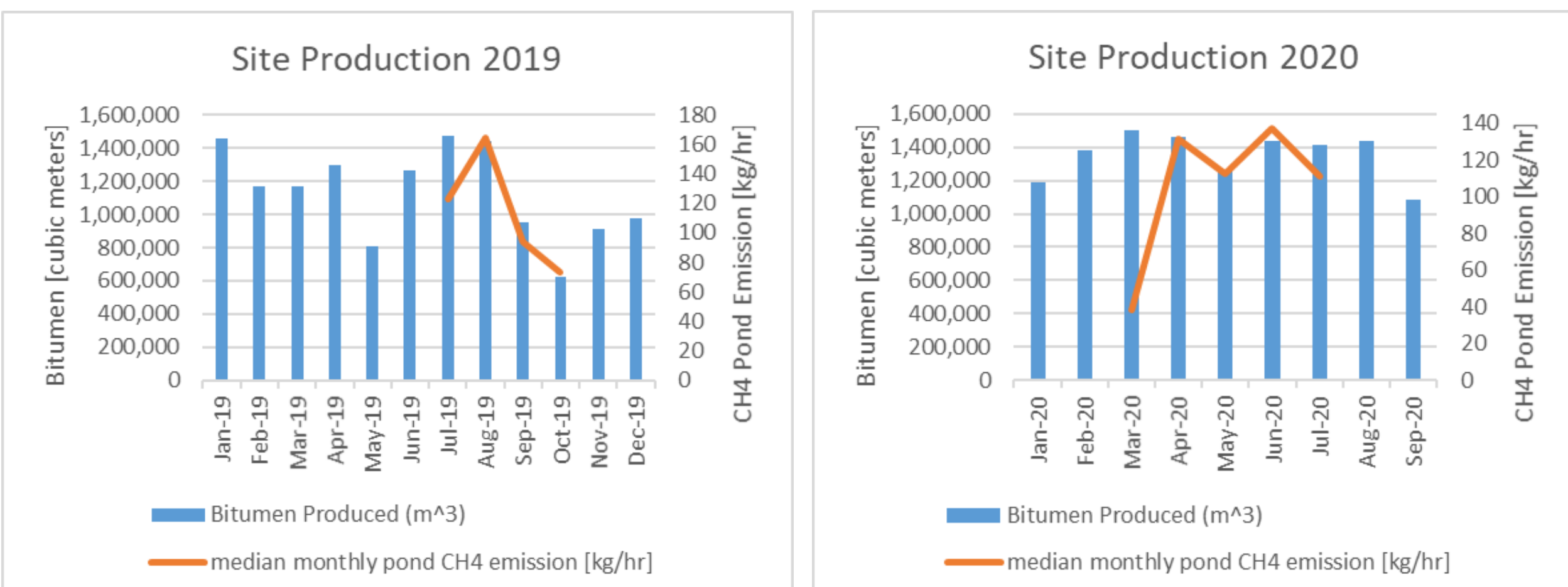

**Figure 12. Bitumen production and GreenLITE™ $CH_4$ tailings pond computed emissions.[13]**

### 5.2. Mine Emissions

$CH_4$ emission rates computed for the mine are shown in Figure 13. Gray data points represent hourly emission rates, and blue data points are the two-day moving average. Known vented emission sources near the northeast boundary of the GreenLITE™ measurement footprint were excluded from these analyses. Average daily emissions were 24.6 metric tons/day during the approximately six-week measurement period. Accounting for the estimated area of the mine pit at the time measurements were taken (3.8 km$^2$), the average daily emission rate scales to an annual flux of 24.5 t/ha/yr. Similar to the temporal variations seen in tailings pond emissions, the variability in estimated mine emissions shown in Figure 13 are modulated by mine activity, the associated localized wind pattern, and atmospheric state. The variability over this six-week period again emphasizes the potential for significant biases in annually reported emissions that are based on short periodic measurements versus a continuous or long-term measurement approach.

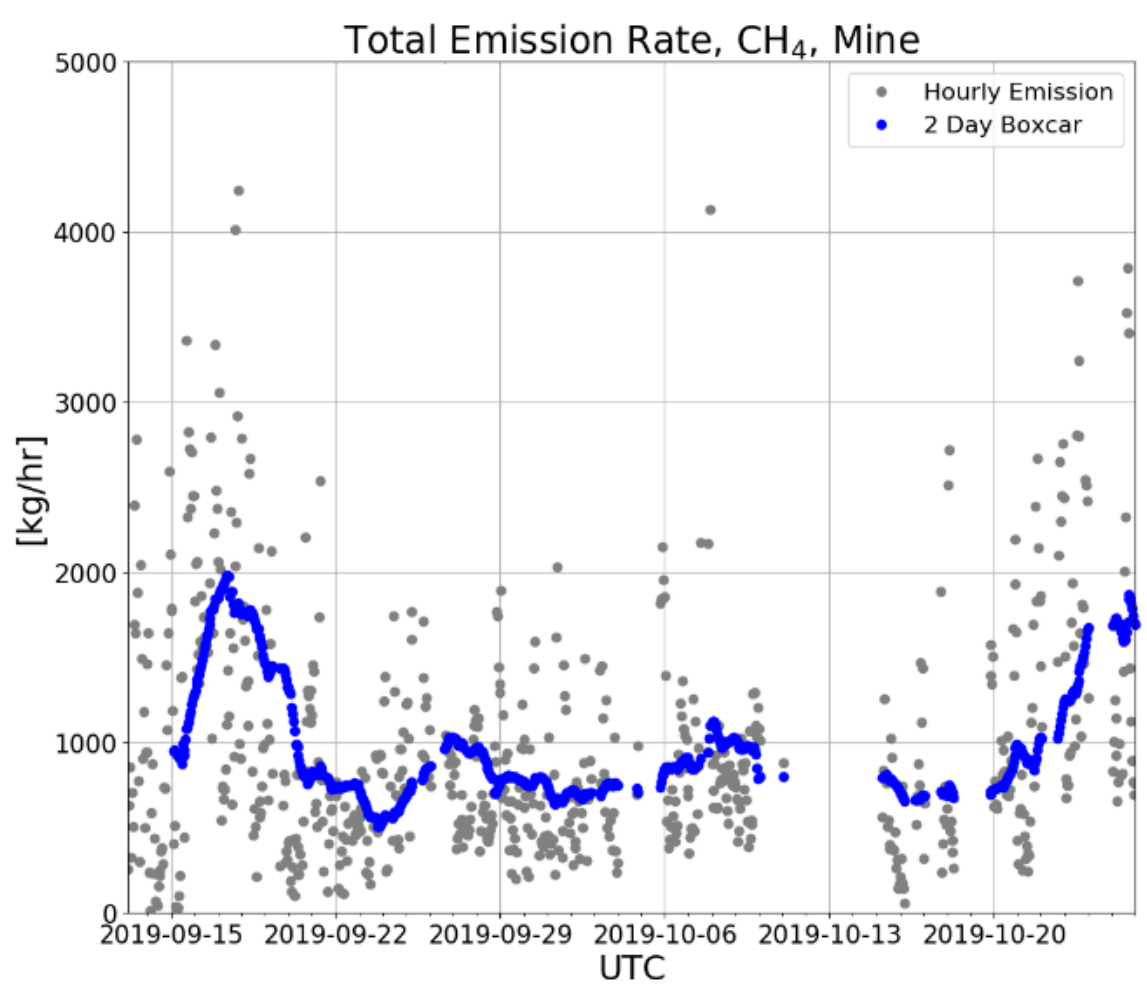

**Figure 13. Open-pit mine methane emissions.**

Like the tailings pond, several $CH_4$ emission studies have been conducted at local mine pits over the last decade using many of the same measurement and emission estimation techniques. Results from these studies are summarized in Table 2. Once

[13] Alberta Energy Regulator, 2021 Statistical Reports ST39 2020, https://www.aer.ca/providing-information/data-and-reports/statistical-reports/st39 last accessed 7/7/2021.

again, the emission values shown in Table 2 provide only a qualitative comparison of emission estimation techniques from multiple seasons and years due to the utilization of measurement footprints that vary significantly.

**Table 2. Historical open-pit mine emission studies [AECOM, 2021].**

| Method | Year | Sampling Period | Mine $CH_4$ Emission (t/yr) |
|---|---|---|---|
| Flux Chamber | 2012 | Late Aug | 10524 |
| | 2013 | Mid Oct | 34684 |
| | 2014 | Early Aug | 22 |
| | 2016 | Aug-Sep | 81 |
| | 2017 | Early Aug | 273 |
| | 2019 | Fall | 33 |
| WindTrax-IDM | 2015 | Sep-Oct | 13391 |
| | 2016 | Aug-Sep | 14746 |
| | 2018 | Apr-May | 9500 |
| | 2019 | Feb-Mar | 11738 |
| | 2019 | Jul-Aug | 12077 |
| | 2019 | Oct-Nov | 13187 |
| CALPUFF-IDM | 2015 | Sep-Oct | 3093 |
| | 2016 | Aug-Sep | 12552 |
| | 2017 | Mid Aug | 12915 |
| | 2018 | Apr-May | 32045 |
| | 2019 | Apr | 14980 |
| | 2019 | Aug | 4664 |
| | 2019 | Sep | 5336 |
| GreenLITE™ | 2019 | Sep-Oct | 8982 |

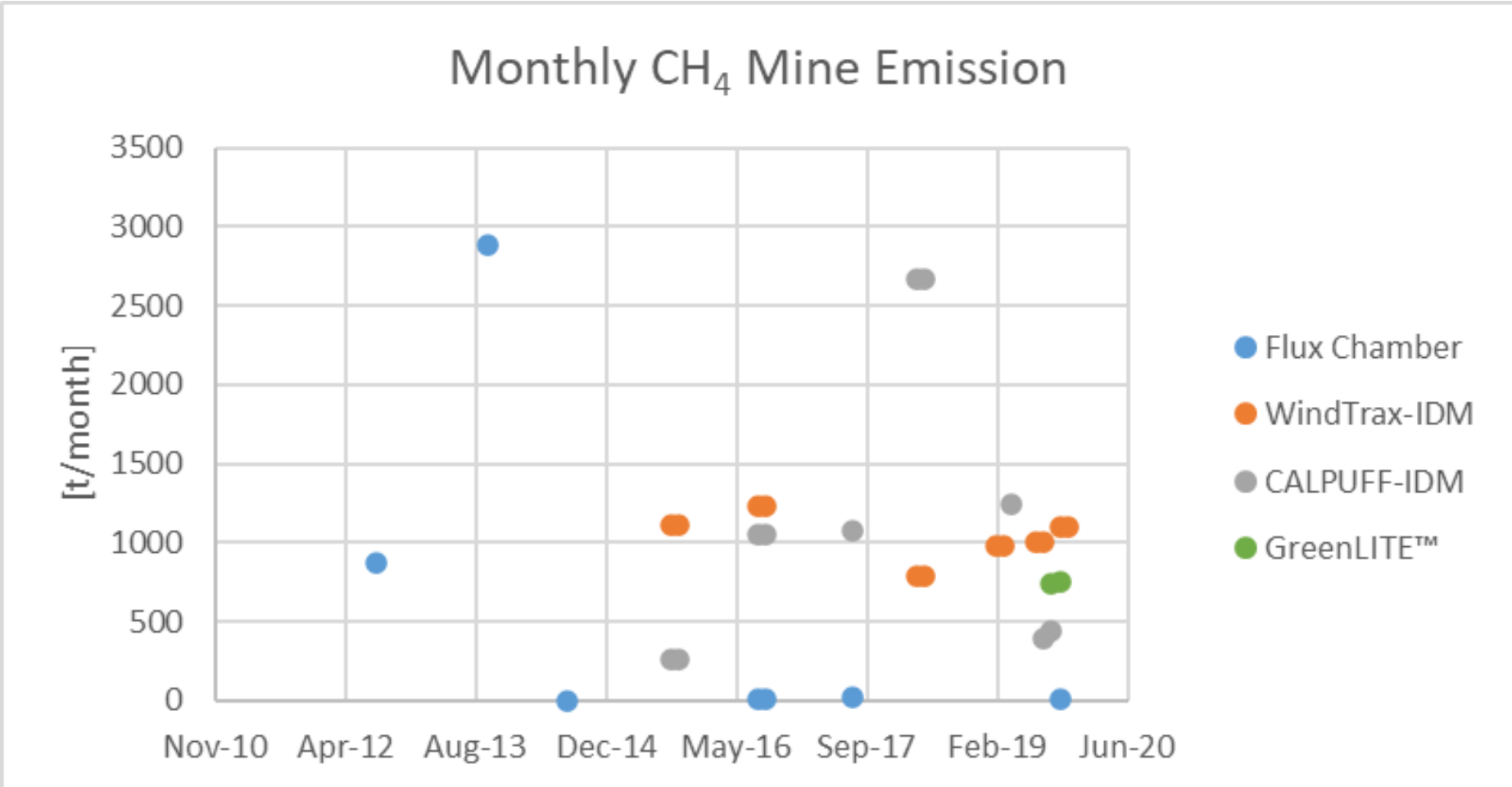

**Figure 14. Historical mine emission studies.**

### 5.3. Uncertainty/Error in Estimates of Emissions

Several factors contribute to uncertainty in estimates of emissions from both the pond and mine environments. Such factors include the accuracy of measured surface meteorology, measured concentrations, and IDM fidelity/user-defined parameters, such as source emitter size/location and input terrain information. Monte-Carlo style simulations were run to quantify the error in retrieved emission rates associated with variability in surface meteorology and instrument measurement precision. Hourly estimates of emissions are achieved by averaging the primary dispersion model input parameters on an hourly basis – namely, measured chord gas concentration, surface air temperature (T), surface air pressure (P), surface air relative humidity (RH), wind direction, and wind speed. Hourly variability in surface air T/P/RH, wind speed, and wind direction were quantified by averaging hourly variability over three separate days during the GreenLITE™ pond measurement campaign in spring of 2020. Of the many days for which wind consistently had a westerly component, which allowed for emissions to be computed for all or most hours of the day, the three days chosen at random were 22 Mar 2020, 27 Mar 2020, and 8 Apr 2020, and the resulting variances are shown in the top portion of Table 3. Previously determined GreenLITE™ instrument measurement precision was used for variability in measured chord concentrations and is provided at the bottom of Table 3. Emissions were retrieved for a given day and hour over ten runs while varying input quantities by uniform random distribution that spanned +/- the average hourly variability (surface meteorology) and instrument measurement precision in Table 3. The variance in resulting emissions over the ten runs represents the error associated with variability of surface meteorology and instrument precision in retrieved hourly emission values, as shown in Table 4.

**Table 3. Meteorological parameter variances and system measurement precision used in Monte-Carlo simulation.**

| Parameter | 1-sigma Average Hourly Variability |
|---|---|
| Surface air pressure | 0.251 mbar |

| | |
|---|---|
| Surface air relative humidity | 1.57% |
| Surface air temperature | 0.262 K |
| Wind speed | 0.437 m/s |
| Wind direction | 7.97° |
| **Parameter** | **System Measurement Precision** |
| $CH_4$ chord concentration | 0.05 ppm |

**Table 4. Emissions error results of Monte-Carlo simulation.**

| UTC | | $CH_4$ [metric ton/day] | |
|---|---|---|---|
| **Date** | **Time** | **10-Run Avg** | **10-Run Std Dev** |
| 21 Mar 2020 | 02:00 – 03:00 | 4.67 | 1.98 |
| 23 Mar 2020 | 08:00 – 09:00 | 5.16 | 1.98 |
| 25 Mar 2020 | 14:00 – 15:00 | 2.10 | 2.93 |
| 6 Apr 2020 | 20:00 – 21:00 | 6.85 | 5.43 |
| 10 Apr 2020 | 02:00 – 03:00 | 1.10 | 2.63 |
| 12 Apr 2020 | 08:00 – 09:00 | 1.49 | 1.81 |
| 15 Apr 2020 | 14:00 – 15:00 | 3.29 | 2.27 |
| 16 Apr 2020 | 20:00 – 21:00 | 8.99 | 4.44 |
| | Avg: | 4.21 | 2.93 |

Dispersion models have intrinsic uncertainties of their own. [Chowdury, 2015] carried out an inert tracer study to characterize the performance of SCICHEM in predicting plume dispersion. In the study, SCICHEM results were compared to plume measurements taken at approximately 2, 4, and 6 km downwind of the tracer release point. In the scenarios of our tailings pond and open-pit mine, measurement chords are directly above the assumed emission sources. Therefore, receptor locations are placed directly over simulated release areas in our SCICHEM simulations. For this reason, the errors reported by [Chowdury, 2015] at 2 km will be referenced. At 2 km, [Chowdury, 2015] reported a normalized mean square error (NMSE) of 2.18% and a normalized mean bias (NMB) of 0.63% in observed versus predicted plume concentration. Based on the estimated emission sensitivity to uncertainty in concentration that was characterized in the aforementioned Monte-Carlo studies, the errors reported by Chowdury correspond to errors in estimated emissions of 5.8e-6 t/day and 1.3e-6 t/day, respectively. The Chowdhury study was conducted under topographical conditions that are assumed to be well defined. Another key difference worth noting between the Chowdhury study and the GreenLITE™ oil sands application is the use of a tracer point source versus modeled extended source. The reported IDM errors associated with modeling a point source can be expected to scale upward for an extended source. Future work will include characterization of IDM uncertainty as used with GreenLITE™ measurements in the oil sands environment.

Global semi-static DEMs are ill-suited to describe the dynamic landscapes of tailings ponds and open-pit mines. The areas of the tailings pond and mine that are considered as sources of emissions comprise a fraction of the total area of terrain passed into and taken into account by SCICHEM – 8% for the pond and 11% for the mine. Still, approximated terrain for the tailings pond and especially the open-pit mine for which SCICHEM simulations were performed may impact the accuracy of dispersion modeling and subsequent emission estimates. Irrespective of the measurement approach used with an IDM, IDMs, like any atmospheric model or emission estimation approach, are inherently less accurate in complex topographies and environments where horizontal homogeneous meteorology cannot be assumed [Flesch, 2005b; Hu, 2016], and in particular for a relatively large depression such as an open-pit mine [Nahian, 2020]. Future work should utilize current topography in dispersion

modeling whenever possible to minimize these errors. Computational fluid dynamic models that characterize the atmosphere in the vicinity of the complex terrain may reduce this error further but may be computationally prohibitive.

The simulated release area size depicted by the white rectangles shown in Figure 3 were chosen to 1) cover the along-chord extent of pond that was assumed to be an emission source and 2) account for a SCICHEM (v3.2) bug that limited the simulated release area to be less than 360 m in one dimension. An idealized simulation would consider the entire pond and west beach as emission sources. For this reason, a study was performed to characterize the relationship between simulated release area size and retrieved emissions. As may be expected, it was found that larger release areas in the cross-wind direction produced larger emission estimates, while larger release areas in the along-wind direction produced smaller emission estimates. Future work should utilize a later version of SCICHEM that allows for release areas to be greater than 360 m in all dimensions, or an alternate IDM, to allow for flexibility in determining and implementing ideal release area sizes, shapes, and locations which will improve the accuracy in estimated emissions.

## 6. Conclusion

A novel approach to estimate fugitive emissions from the mine pit and tailings pond of a large oil sands operation has been demonstrated that utilizes the GreenLITE™ gas measurement system and the SCICHEM IDM. $CH_4$ emissions from a tailings pond were estimated to be 7.2 t/day for Jul-Oct 2019, and 5.1 t/day for Mar-Jul 2020. $CH_4$ emissions from the mine pit were estimated to be 24.6 t/day for Sep-Oct 2019. Estimated emission rates for both the tailings pond and mine are in family with several recent studies at the oil sands site that employed a variety of measurement and emission estimation approaches. Emissions from wide area sources, such as oil sands tailings pond and open-pit mines, tend to vary both spatially and temporally. For the purposes of 1) emission regulation reporting/compliance and 2) emission mitigation planning, implementation, and assessment, an ideal measurement solution would include continuous measurement over extended time periods and cover an area of interest with spatial resolution high enough to identify and apportion emissions to specific sectors within the measurement footprint. The continuous, integrated-path, wide-area coverage of the GreenLITE™ system was used to estimate and apportion $CH_4$ emission at the open-pit mine as implemented in the two-transceiver configuration, which allows for 2-D mapping. While 2-D mapping is not possible in the one-transceiver configuration employed at the tailings pond, apportionment of emissions is possible to a lesser degree and improves with the number of measurement chords passing over the assumed emission source.

The approach demonstrated here may be applicable to a variety of wide-area emission scenarios, to include oil and gas production, wastewater treatment plants, landfills, feedlots, wetlands, permafrost, cities, and shipping ports. Future work may involve comparisons of emissions results using additional, alternative IDMs, and should incorporate current topography in dispersion modeling whenever possible. Furthermore, a flux-gradient approach may be explored in a future GreenLITE™

deployment utilizing concentration measurements at multiple heights. Such an approach could reduce the computational expense associated with the IDM method.

## Disclaimer

The GreenLITE™ gas measurement system has been co-developed by Atmospheric and Environmental Research, Inc (AER), and Spectral Sensor Solutions, LLC (S3). Timothy G. Pernini and T. Scott Zaccheo are employees of AER. Jeremy T. Dobler and Nathan Blume are employees of S3.

## Acknowledgements

We would like to acknowledge Emissions Reduction Alberta, the Canada's Oil Sands Innovation Alliance (COSIA) industrial partners Imperial Oil, for initial introduction and support for this project, and Canadian Natural Resources Limited (CNRL) for their significant support extending this work to include the mine and spring thaw deployments. We would also like to acknowledge the CNRL Horizon staff for excellent onsite logistics support required to execute this work.

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
