# Peer review of "Estimating Oil Sands Emissions using Horizontal Path-Integrated Column Measurements"

_Atmospheric Measurement Techniques, 2021_

## Referee Comment (RC2)

Pernini et al. presents the novel open-path laser system "GreenLITE™" with multiple reflectors and transceivers deployed in a tailings pond and an open-pit mine experiments. This system has a much wider spatial coverage that might provide valuable information to estimate the fugitive GHG emissions in addition to other measurement techniques. This is a clearly written manuscript with methods and results arranged in a logical order, which made it easy to follow. Overall, this work is appropriately placed in AMT. The content of the paper, which covers GHG measurements and its application to quantify the fugitive $CH_4$ emissions together with an inverse dispersion model, is relevant to the journal and its readers. Much of the analyses and conclusions are sound. I recommend publication after the following comments are addressed.

**General comments:**

The results of this paper would be much stronger if it places the emission estimates together with better uncertainty quantification. The authors have already touched on this issue in section 5.3:

First, Table 4 shows that the impact of surface meteorology and measurement precision is significant and not negligible. However, the last sentence in the first paragraph of section 5.3 only pointed out these errors without further analysis and discussion. For example, what are the potential reasons for the day-to-day & the day-night differences? what are the impacts of these uncertainties on the total emission estimates?

Second, the authors have discussed different sources of uncertainties and some potential improvements. However, it remains unclear to me which one might have a major impact than the others. It would be interesting to see somewhere that explores the impact rankings of these sources of uncertainties. For example, if using a different dispersion model looked at the same thing, would the authors get a similar magnitude of the emission estimates?

The authors do not necessarily need a major modification given that the limitations of the approach are discussed in the end, but I just wanted to caution about overconfidence in retrieved emissions. Therefore, it would be better to add proper uncertainty estimates together with the absolute emissions both in the abstract (P1 Line19) and the main text (P11 Line 8 and P23 Line16).

General note on figures and tables:

Much effort should be put to avoid the redundancy of the figures and tables. For example, Figure 1 and Figure 3 could be combined; Figure 7 and 8 could go to supplement; Table1 and Figure11, Table 2 and Figure 14 are redundant.

**Specific comments:**

P8 Line 15: It would be interesting to add the measured concentrations for each chord either in Figure 10 or a supplement figure. This might to some extent be helpful in 1) verifying whether the definition of the background field is appropriate; 2) providing information on the spatial variations of the concentration and emissions; 3) understanding what the contribution from each individual chord to the total amount is.

P9 Line 6: The P23 Line4 describes well the arbitrary definition of the size of these "white rectangles" as well as its limitation, but I don't think I would have understood here, especially for the choice of the width, until reading section 5.3. Maybe a few sentences in terms of this model setup should come earlier.

P11 Line 8: Add an uncertainty for the period Jul-Oct 2019. The superscript "6" is not an exponential notation here, it might be good to move it to the end of the sentence to avoid misunderstanding.

P11 Line 10: Given the seasonal variability of the emissions, it would be good to explain how to scale these emission rates over the sampling period to annual fluxes.

Figure 6: What are the reasons that may explain the negative values? How are these negative values accounted for in the emission estimates?

Figure 9: The caption and the main text are "two-day moving average", the legends for the blue dots are inconsistent: "1 Day Boxcar" in Figure 9, while "2 Day Boxcar" in Figure 6 and Figure 13.

Figure 12: Eliminate the day in the x-label.

P18 Line 4: The method and justification for discarding these known emissions sources need to be given. By using the lowest concentration within the mine domain, how did the authors exclude the impact of these nearby sources?

P22 Line 4: It might be insufficient to use the Chowdury 2015 study as a reference to quantify the uncertainty in the dispersion model. As already noted by the authors, there are many differences in terms of the model setup.

Font sizes in some figures could be larger, e.g., Figure 10.

Units should be checked in many places throughout the article to improve clarity and comparability, e.g., t/day in abstract, metric tons/day in P11 Line 8, metric ton/day in Table 4, t/yr in Table 1, t/month in Figure 14.

---

## Author Response (AR1)

AMT-2021-230
Submitted on 27 Jul 2021 | Research article

Estimating Oil Sands Emissions using Horizontal Path-Integrated Column Measurements

Timothy G. Pernini, T. Scott Zaccheo, Jeremy T. Dobler, and Nathan Blume
* * *
**RC1**: 'Comment on amt-2021-230', Anonymous Referee #1, 02 Nov 2021

**General Comments:**

This manuscript describes the novel application of a scanning open-path laser absorption system to the real-world problem of fugitive methane emissions from Alberta oil sands tailings ponds and mine faces. The 2D methane concentration fields obtained with the laser system are used in conjunction with an inverse dispersion model to derive emission estimates. The advantages of this approach regarding the establishment of continuous spatially representative time series of emission rates is demonstrated, particularly through the quantitative characterization of the spring pulse in the methane flux from a pond as the ice breaks, something that has long been speculated to be of potential importance.
A comprehensive comparison of the open-path laser based results with previously published fluxes is presented and convincingly illustrates some of the issues with using snap-shot flux chamber measurements to try to quantify emissions representative of a large, heterogeneous surface source. Of course the presented method is not without issues; as explained below, there are reasons to believe that the derived emission rates are more likely to be under- rather than over-estimates. Nevertheless, methods such as the one described in this manuscript are approaching maturity and will hopefully help to modernize industrial emissions reporting protocols in the coming years.
The manuscript is well written and can be published after minor modifications. Details on suggested improvements are given below.

Authors' response: Thank you for the constructive review comments. Please see below for a summary of how each specific comment has been addressed.

**Specific Comments:**

Page 5 Line 9: specify the period of time
Authors' response: A representative range of instrument integration times has been added to text.

P10L19: Using the lowest concentration inside your relatively small domain in close proximity to strong sources may not give you a proper background concentration. Overestimating the background concentrations will result in an underestimate of the emissions. Can you add some evidence that your background concentrations are low enough (by comparison with time series from other measurements, for example)?
Authors' response: Background for purposes of computing emissions with the GreenLITE™ system and measurement set up is not necessarily "ambient" local background. We want background directly upwind of our measurement footprint, which may take into account potentially strong local sources that are slightly farther upwind. We want this because our measurement paths will measure both true ambient background and anything else blowing across our footprint (in addition to any site emissions). Ideally we'd have an integrated path measurement directly upwind of measurement footprint (as we do with the background measurement paths at the pond), or a set of point measurements directly upwind that span the diameter of the measurement footprint. Text has been added to describe some of these points.

P14L10: As I see it, it is an indication that the diurnal cycles of the concentrations are due to a common (probably meteorological) driver, but I'm not sure you can deduce that there is no bias due to local sources.
Authors' response: Text has been removed.

P22L12: these errors are so tiny compared to most of the other uncertainties involved that they're not really worth discussing in this much detail – unless you can separately quantify and discuss other errors (e.g. background concentrations, winds, non-stationarity of meteorological conditions) in similar detail.
Authors' response: Text has been removed and simplified.

P23L1: I think the DEM / height above ground issue may be significant and worth a bit of extra effort. If the depth of the mine (below the measurement level) is underestimated, then the wind speeds may also be underestimated, and by extension the same will be true for the dispersion / dilution of the emitted methane, resulting in an underestimate of the emissions. Perhaps the fastest way to get a rough quantitative estimate of the magnitude of the difference would be a Monte-Carlo approach of varying the elevations of the beam paths above ground, to simulate the mine being deeper than given by the DEM. Also worth exploring might be the dependence on surface roughness ($z_0$).
Authors' response: We intend to pursue the suggestion for a Monte Carlo simulation to characterize the impact of mine depth on emission retrievals in future work. Thank you for the suggestion. We have added text to address this, as well as a reference to a recent paper that discusses the effect of mine depth on atmospheric transport.

Figures and Tables: there are more figures than necessary, with some redundancy that can easily be fixed. A few figures, and all the tables, should be moved to a Supplement to improve the flow.
Author's response: Several tables and figures have been moved to an appendix section.

Specific suggestions:

Figure 3 is mostly a repeat of Figure 1. Combine these two, and mark the center release points.
Authors' response: Figure 1 has been replaced with figure 3. Figure 3 has been deleted.

Figure 4 currently appears in the text before Figure 3; relabel.
Authors' response: Figure labelling has been updated.

Utilize the space in Figure 6 more efficiently by eliminating the large white space in the middle; use a broken x-axis.
Authors' response: Emissions are now plotted separately for the two measurement campaigns.

Insert the data in Figure 8 into Figure 9; this will make it easier for the reader to line up the melt, and it reduces the figure count. Mark the April 28 peak on the graph for easier identification.
Authors' response: Figure 8 has been moved to the Appendix. Figure 9 has been removed and incorporated into Figure 6.

Eliminate Figure 7; redundant.
Authors' response: This figure highlights two important points in time: 1. When the daily high temperature reached above freezing during the second week of April, and 2. Daily average temperature

above freezing during third week of April. During both times we saw an increase in emissions. Figure 7 has been moved to the appendix.

Figure 10: mark the daylight / nighttime difference by shading.
Authors' response: Day/night time shading has been added to figure.

Figure 12: the correlation is surprisingly good! You can add error bars (interquartile range or so) to indicate the high uncertainty associated with the first monthly median.
Authors' response: Thank you for the suggestion. Our prior statistical analysis for these monthly emission data sets indicated that the under sampling of months does not manifest itself as a larger spread (higher variance) in the data. Rather, we suspect that the under sampling has caused a bias in the monthly median emission. Since high emission outliers cause these monthly data sets to have a skewed distribution, including error bars would alter the emission axis scale such that the intended correlation between bitumen production and pond emission would be difficult to observe.

All tables should go into the supplement. The information in Tables 1 and 2 is already present in Figs. 11 and 14.
Authors' response:  Tables 1 and 2 have been moved to the appendix.

**Technical Corrections:**

Throughout, references still need to be standardized (currently a mixture of footnotes and citations) and completed (e.g. "et al" if multiple authors).
Authors' response: Reference have been standardized.
* * *
**RC2**: 'Comment on amt-2021-230', Anonymous Referee #2, 15 Nov 2021

Pernini et al. presents the novel open-path laser system "GreenLITE™" with multiple reflectors and transceivers deployed in a tailings pond and an open-pit mine experiments. This system has a much wider spatial coverage that might provide valuable information to estimate the fugitive GHG emissions in addition to other measurement techniques. This is a clearly written manuscript with methods and results arranged in a logical order, which made it easy to follow. Overall, this work is appropriately placed in AMT. The content of the paper, which covers GHG measurements and its application to quantify the fugitive CH4 emissions together with an inverse dispersion model, is relevant to the journal and its readers. Much of the analyses and conclusions are sound. I recommend publication after the following comments are addressed.

Authors' response: Thank you for the constructive review comments. Please see below for a summary of how each specific comment has been addressed.

**General comments:**
The results of this paper would be much stronger if it places the emission estimates together with better uncertainty quantification. The authors have already touched on this issue in section 5.3:

First, Table 4 shows that the impact of surface meteorology and measurement precision is significant and not negligible. However, the last sentence in the first paragraph of section 5.3 only pointed out

these errors without further analysis and discussion. For example, what are the potential reasons for the day-to-day & the day-night differences? what are the impacts of these uncertainties on the total emission estimates?

Authors' response: At times when weather (T/RH/P/wind) is changing most rapidly (weather fronts, sun rise/set, etc), the impact of averaging these weather parameters over an hour may introduce additional error in emissions versus times when weather is stable. As for impact on total emission estimates, that's the intent of the Monte-Carlo error analysis. Text has been added to section 5.3 to address this comment.

Second, the authors have discussed different sources of uncertainties and some potential improvements. However, it remains unclear to me which one might have a major impact than the others. It would be interesting to see somewhere that explores the impact rankings of these sources of uncertainties. For example, if using a different dispersion model looked at the same thing, would the authors get a similar magnitude of the emission estimates?

Authors' response: Thank you for the suggestion. We will explore use of alternative dispersion models in future work, as well as a more complete and inclusive uncertainty analysis. Section 6 states our intent to explore alternative IDMs in future work.

The authors do not necessarily need a major modification given that the limitations of the approach are discussed in the end, but I just wanted to caution about overconfidence in retrieved emissions. Therefore, it would be better to add proper uncertainty estimates together with the absolute emissions both in the abstract (P1 Line19) and the main text (P11 Line 8 and P23 Line16).

Authors' response: Results of the error analysis run for the 2020 campaign at the pond have been added to the abstract and conclusion section 6.

General note on figures and tables:
Much effort should be put to avoid the redundancy of the figures and tables. For example, Figure 1 and Figure 3 could be combined; Figure 7 and 8 could go to supplement; Table1 and Figure11, Table 2 and Figure 14 are redundant.

Authors' response: Figures 1 and 3 have been combined. Figures 7 and 8 have been moved to the appendix. Tables 1 and 2 have been moved to the appendix.

**Specific comments:**

P8 Line 15: It would be interesting to add the measured concentrations for each chord either in Figure 10 or a supplement figure. This might to some extent be helpful in 1) verifying whether the definition of the background field is appropriate; 2) providing information on the spatial variations of the concentration and emissions; 3) understanding what the contribution from each individual chord to the total amount is.

Authors' response: Thank you for the suggestion. Both measured concentration and retrieved emission on a per-chord basis have been analyzed and used to apportion emissions to pond sectors. Results will be presented in a future work.

P9 Line 6: The P23 Line4 describes well the arbitrary definition of the size of these "white rectangles" as well as its limitation, but I don't think I would have understood here, especially for the choice of the width, until reading section 5.3. Maybe a few sentences in terms of this model setup should come earlier.

Authors' response: Text has been added to section 4 to describe the simulated release areas.

P11 Line 8: Add an uncertainty for the period Jul-Oct 2019. The superscript "6" is not an exponential notation here, it might be good to move it to the end of the sentence to avoid misunderstanding.
Authors' response: A preliminary uncertainty analysis was preformed using a few days' worth of data during the 2020 measurement period. Text has been added to section 5.3 do describe future intentions for a more comprehensive uncertainty analysis. The superscript "6" has been relocated.

P11 Line 10: Given the seasonal variability of the emissions, it would be good to explain how to scale these emission rates over the sampling period to annual fluxes.
Authors' response: Text has been added to section 5.1 to explain the purpose of extrapolating emission to annual values, and why this could add significant uncertainty.

Figure 6: What are the reasons that may explain the negative values? How are these negative values accounted for in the emission estimates?
Authors' response: Text has been added to section 5.1 that discusses negative emission values.

Figure 9: The caption and the main text are "two-day moving average", the legends for the blue dots are inconsistent: "1 Day Boxcar" in Figure 9, while "2 Day Boxcar" in Figure 6 and Figure 13.
Authors' response: Plots and text have been updated for consistency.

Figure 12: Eliminate the day in the x-label.
Authors' response: We assume the reviewer is suggesting removing the two-digit year from the x-labels. This has been done.

P18 Line 4: The method and justification for discarding these known emissions sources need to be given. By using the lowest concentration within the mine domain, how did the authors exclude the impact of these nearby sources?
Authors' response: Text has been added to section 5.2 to describe how known local sources were excluded.

P22 Line 4: It might be insufficient to use the Chowdury 2015 study as a reference to quantify the uncertainty in the dispersion model. As already noted by the authors, there are many differences in terms of the model setup.
Authors' response: The text describing the Chowdury 2015 has been updated and simplified.

Font sizes in some figures could be larger, e.g., Figure 10.
Authors' response: Figure font sizes have been increased.

Units should be checked in many places throughout the article to improve clarity and comparability, e.g., t/day in abstract, metric tons/day in P11 Line 8, metric ton/day in Table 4, t/yr in Table 1, t/month in Figure 14.
Authors' response: "metric" has been added to units in the paper that refer to tons for consistency. Tables 1 and 2 (now in appendix) contain values and units (t/yr) as reported in other studies. For qualitative comparison, the emissions from our oil sands study have been included in these tables in these same units. Figure 11 has been updated from t/yr to t/month to be consistent with figure 14, which are intended to show potential historical trends.

---

## Author Response (AR2)

AMT-2021-230
Submitted on 27 Jul 2021 | Research article
Estimating Oil Sands Emissions using Horizontal Path-Integrated Column Measurements
Timothy G. Pernini, T. Scott Zaccheo, Jeremy T. Dobler, and Nathan Blume
* * *
13 Dec 2021
Associate Editor decision: Publish subject to technical corrections
by Reem Hannun
**Comments to the author**:
A minor comment: It is still unclear whether the error estimation applies to only the tailings ponds emissions or both the tailings pond and the mine pit. If just the former, why are there no estimates for the mine pit?

Non-public comments to the Author:
If you feel it is appropriate, you could consider moving the appendix to a supplement instead.

**Author's response:**
Thank you for the comments (above) regarding our paper submission.

Section 5.3 describes our approach for a preliminary error estimation which utilized measurements from the Jul-Oct 2019 time period. We have fixed the abstract text by associating the 2.9 metric tons/day error estimate with the Jul-Oct 2019 time period to be consistent with Section 5.3 and the text in the Conclusion Section 6 (line 16). Lastly, section 5.3 (lines 21-22) state our intention to formalize our error analysis and run during each future measurement campaign.

The content currently included in our Appendix A seemed to fit the AMT description for an appendix versus a supplement (https://www.atmospheric-measurement-techniques.net/submission.html). If AMT feels this content is better suited for a supplement, we will gladly transfer this content to a supplement. Please advise.